# Adjusting 1.5 degree C climate change mitigation pathways in light of adverse new information

Ajay Gambhir [1] ✉, Shivika Mittal [1], Robin D. Lamboll [2], Neil Grant[1,3], Dan Bernie [4,5], Laila Gohar[4], Adam Hawkes [6], Alexandre Köberle [1,7,8,9], Joeri Rogelj [1,2,10] & Jason A. Lowe[4,11]

Understanding how 1.5 °C pathways could adjust in light of new adverse information, such as a reduced 1.5 °C carbon budget, or slower-than-expected low-carbon technology deployment, is critical for planning resilient pathways. We use an integrated assessment model to explore potential pathway adjustments starting in 2025 and 2030, following the arrival of new information. The 1.5 °C target remains achievable in the model, in light of some adverse information, provided a broad portfolio of technologies and measures is still available. If multiple pieces of adverse information arrive simultaneously, average annual emissions reductions near 3 $GtCO_2$/yr for the first five years following the pathway adjustment, compared to 2 $GtCO_2$/yr in 2020 when the Covid-19 pandemic began. Moreover, in these scenarios of multiple simultaneous adverse information, by 2050 mitigation costs are 4-5 times as high as a no adverse information scenario, highlighting the criticality of developing a wide range of mitigation options, including energy demand reduction options.

The remaining Paris Agreement-compliant carbon budget is small (at 500 $GtCO_2$ from the start of 2020 for a 50% likelihood warming limit of 1.5 °C), and is quickly being consumed by current emissions (with global carbon dioxide emissions at ~40 $GtCO_2$/yr in 2021[1]). Many modelled scenarios that are still compatible with limiting warming below 1.5 °C exist. However, these predominently cost optimum pathways require rapid reductions in $CO_2$ and other greenhouse gas (GHG) emissions, at 43% below 2019 levels by 2030 (34−60% 5th−90th percentile range), to achieve >50% likelihood of below 1.5 °C warming by 2100, with no or limited temperature overshoot[2]. These pathways include fast transitions from fossil fuels to low-carbon energy, improved energy efficiency and energy demand side measures, and most involve the rapid and large-scale deployment of carbon dioxide removal (CDR) technologies[2].

Several uncertainties remain, which could influence mitigation pathways and choices. Such uncertainties will include geophysical information, which constrains the size of a temperature-consistent global carbon budget; technological information around the cost reduction, scale-up and business case potential of existing and new low-carbon technologies such as renewables, carbon capture and storage (CCS) and carbon dioxide removal (CDR); biophysical information around the potential for land-intensive resources such as bioenergy and afforestation; and behavioural information around the extent to which the world will develop along an energy-efficient or energy-intensive pathway. Some uncertainties will be reduced or resolved in the coming years, and robust mitigation pathway planning should be ready to respond to this information, so that pathways may be adjusted accordingly.

[1]Grantham Institute, Imperial College London, London, UK. [2]Centre for Environmental Policy, Imperial College London, London, UK. [3]Climate Analytics, Berlin, Germany. [4]Met Office Hadley Centre, Exeter, UK. [5]Bristol Medical School, University of Bristol, Bristol, UK. [6]Department of Chemical Engineering, Imperial College London, London, UK. [7]Centre for Climate Finance and Investment, Imperial College Business School, London, UK. [8]Potsdam Institute for Climate Impact Research, Potsdam, Germany. [9]Instituto Dom Luiz, Faculdade de Ciências, Universidade de Lisboa, Lisbon, Portugal. [10]Energy, Climate and Environment Program, International Institute for Applied Systems Analysis, Laxenburg, Austria. [11]Priestley Centre, University of Leeds, Leeds, UK. ✉e-mail: a.gambhir@imperial.ac.uk

A number of studies have used detailed process-integrated assessment models (DP-IAMs) to explore low-carbon pathways, using scenarios in which actions change at some point along the way. For example, several IAM scenarios have included relatively weak levels of mitigation action in the near-term, followed by a sudden ramp-up to meet carbon budget or temperature objectives. Such delays in mitigative action have been shown to impact the feasibility of meeting climate targets, finding that if ambitious mitigation is delayed globally[3–6], or there is fragmented action with certain regions acceding to a global regime later[7,8], the feasibility is substantially reduced, whilst mitigation, when it does occur, is faster and costlier. Many scenarios have included analysis of technological limits, including no new nuclear, CCS deployment failure, limited renewables deployment and restricted biomass availability[9], again finding the need for greater near-term action[7,10], higher costs and reduced feasibility for those scenarios which can still achieve long-term goals[5,11]. Specific analysis on CDR availability has shown that accelerating near-term action is a robust strategy to reduce reliance on CDR[12–15], predominantly driven by accelerated renewables deployment[13]. Reductions in energy demand can both compensate for deployment failure in CDR[16] and increase the feasibility window even in the event of delayed action[5].

These investigations in the literature are of considerable value, but do not explicitly consider the role of mitigation pathway adjustments in responding to adverse information in an adaptive manner. One exception uses a Benefit-Cost IAM (BC-IAM), DICE, to explore a dynamic adaptive mitigation response to continuously updated information on temperature changes resulting from emissions. It finds that such adaptive strategies prevent over-spending on mitigation in low climate sensitivity futures, and under-spending in higher sensitivity futures[17]. A similar approach updates and divides up the remaining carbon budget continuously and spends it smoothly, reducing the chance of overshoot[18]. Another relevant study assessing mitigation under uncertainty[7], now a decade old, explores the different actions and pathways, as well as their costs and feasibility, that could keep the world on track to 1.5–2 °C climate targets, in light of a range of (at the time, projected) 2020 emissions levels.

Here we explore how (initially cost-optimised, perfect-foresight) mitigation pathways might be adjusted in light of adverse information being received at different future time-points, specifically in 2023 and 2028, to tie into the two forthcoming Global Stocktakes (GSTs)[19] of the Paris Agreement. Rather than taking a purely back-casting approach (which develops a single pathway towards a desirable future from a current starting point[20,21]), we instead develop a range of pathways that adapt to new learning from the GSTs, which is assumed to take effect a year after the end of each GST, via a re-optimisation of the mitigation pathway from those points. The new information includes a shrinking of the 1.5 °C-consistent carbon budget[22] in light of updated assessments[23]; reduced expected future growth rates for wind and solar in light of evidence from previous technology pathway deployment growth patterns[24]; reduced future growth rates for CCS and CDR from bioenergy with CCS (BECCS) as well as direct air capture (DAC); expectations that the agriculture, forestry and land use (AFOLU) sector will become a net source, rather than sink, of $CO_2$ emissions over the course of the 21st century. We also explore the consequences of implementing lower energy demand policies and actions in scenarios where all adverse information is received together (see Table 1 for full scenario set). Our analysis sits within a "Dynamic Adaptive Policy Pathways" approach[25], whereby policies and actions are developed to be robust to a range of possible futures (Fig. 1), rather than optimal for a best-estimate future. We use an integrated assessment model (TIAM-Grantham)[26,27] to represent the energy system and related $CO_2$ emissions, a model simulating AFOLU emissions(MAgPIE[28]), an infilling multi-gas model (Silicone[29]) to estimate other greenhouse gases and a probabilistic simple climate model (FaIR[30]), to explore the global mean temperature change profiles that result from our (primarily energy and industrial system $CO_2$-focused) mitigation pathways. The Methods section describes our modelling approach in further detail.

## Results

Table 1 provides a summary and description of the scenarios. In each scenario we simulate a situation in which the (previously cost-optimised) mitigation pathway pauses at 2025 and 2030, respectively. From these time points, and with the previous years' energy system changes now "locked in", a new cost-optimal mitigation pathway, encompassing new information on the revised carbon budget, revised AFOLU net emissions, revised technology maximum deployment growth rates, and future levels of energy demand, are accounted for. This "re-optimisation" thus allows a re-examination and adjustment of

**Table 1 | Set of scenarios aimed at achieving a 1.5 °C target**

| Scenario name | Details |
|---|---|
| 1p5_ref | 1.5 °C-consistent carbon budget of 500 GtCO$_2$ (from 2020) is targeted, with no new adverse information received |
| 1p5_lowBudget_25 | Decision to aim for 400 GtCO$_2$ carbon budget (still starting from 2020), from 2025 onwards, following 2023 GST |
| 1p5_limCCS_25 | Realisation that CCS can only scale at a maximum of 3% per year, not initially-envisaged 20%, with new constraint applied from 2025. |
| 1p5_limDAC_25 | Realisation that DAC can only scale at a maximum of 3% per year, not initially-envisaged 12.5%, with new constraint applied from 2025. |
| 1p5_limRNW_25 | Realisation that solar PV and wind can only scale at a maximum of 6% and 5% per year respectively, not initially-envisaged 30% and 10%, with new constraint applied from 2025. |
| 1p5_limAFOLU_25 | Realisation that AFOLU will contribute cumulative positive emissions of 46 GtCO$_2$ rather than net removals of 135 GtCO$_2$ over period 2020–2100, with new constraint applied from 2025. |
| 1p5_limBundle_25 | Adverse information on carbon budget, CCS, DAC and renewables (as detailed above) received in 2023, with constraints applied from 2025 |
| 1p5_limBundleLD_25 | Adverse information on carbon budget, CCS, DAC and renewables (as detailed above) received in 2023, with constraints applied from 2025, and with low energy demand assumptions applied from 2025. |
| 1p5_lowBuget_30 | Decision to aim for 400 GtCO$_2$ carbon budget (still starting from 2020), from 2030 onwards, following 2028 GST |
| 1p5_limCCS_30 | Realisation that CCS can only scale at a maximum of 3% per year, not initially-envisaged 20%, with new constraint applied from 2030. |
| 1p5_limDAC_30 | Realisation that DAC can only scale at a maximum of 3% per year, not initially-envisaged 12.5%, with new constraint applied from 2030. |
| 1p5_limRNW_30 | Realisation that solar PV and wind can only scale at a maximum of 6% and 5% per year respectively, not initially-envisaged 30% and 10%, with new constraint applied from 2030. |
| 1p5_limAFOLU_30 | Realisation that AFOLU will contribute cumulative positive emissions of 46 GtCO$_2$ rather than net removals of 136 GtCO$_2$ over period 2020–2100, with new constraint applied from 2030. |
| 1p5_limBundle_30 | Adverse information on carbon budget, CCS, DAC and renewables (as detailed above) received in 2028, with constraints applied from 2030 |
| 1p5_limBundleLD_30 | Adverse information on carbon budget, CCS, DAC and renewables (as detailed above) received in 2028, with constraints applied from 2030, and with low energy demand assumptions applied from 2030. |

the cost-optimal mitigation pathway in light of new information. But there is no opportunity for hindsight in which the initially embarked-upon mitigation pathway can be revisited from scratch – thereby reflecting the reality of mitigation choices made at any time point.

### Emissions and temperature pathways

Emissions pathways (focusing on fossil fuel and industrial $CO_2$) for each of the scenarios begin to diverge in 2025 and 2030, depending on the pathway adjustment timing (Fig. 2), revealing a number of notable features. First, where the available carbon budget is reduced from 500 $GtCO_2$ to 400 $GtCO_2$ from 2025 ("1p5_lowBudget_25"), this leads to a sustained reduction in emissions (of between 0.9 and 2.0 $GtCO_2$/yr) to 2100, compared to the 1.5 °C reference scenario ("1p_ref") in which no new information arrives. The later pathway adjustment (and

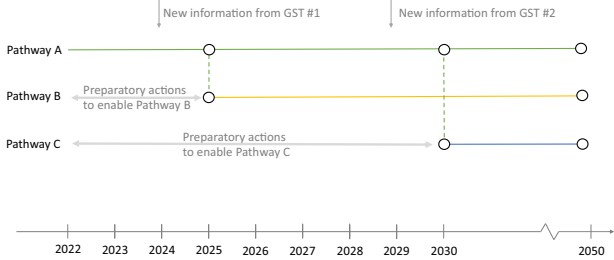

**Fig. 1 | Illustration of three pathways to decarbonisation by 2050.** Pathway 1 (solid green line) represents an optimal pathway based on current knowledge, in which no new information is received. In light of receipt of new information in Global Stocktake (GST1) at end of 2023, pathway shifts to Pathway B in 2025, provided this is possible in light of preparatory actions before 2025. Similarly, in light of new information in GST2 (end of 2028), pathway shifts to Pathway C, provided preparatory actions enable this.

"1p5_lowBudget_30") results in a similar reduction (0.8–2.3 $GtCO_2$/yr) after 2030. A similar pattern (though with more marked emissions reductions of 1.5–3.4 $GtCO_2$/yr over the period 2030–2100) is observed where emissions removals from AFOLU are limited and the pathway adjusted from 2025, equivalent to shrinking the fossil fuel and industrial $CO_2$ carbon budget by just over 180 $GtCO_2$ (36% of the original carbon budget).

Second, limits to CCS and DAC growth rates result in faster near-term mitigation, and less reliance on negative emissions later in the century. In scenarios without DAC and CCS growth limits, net negative emissions are 11.6–14.2 $GtCO_2$ in 2100, versus 6.3 $GtCO_2$ in 2100 when DAC growth is limited from 2025, and 6.6 $GtCO_2$ in 2100 when DAC growth is limited from 2030. When CCS growth is limited from 2025, there is 8.4 $GtCO_2$ of net negative emissions by 2100, and 8.7 $GtCO_2$ by 2100 when CCS growth is limited from 2030. This is because CCS deployment growth limits negative emissions via BECCS, throughout the period to 2100.

Third, limits to the growth of wind and solar from 2030 onwards do not lead to a perceptible change in the emissions pathway compared to the 1.5 °C reference scenario, since (as explained below) other low-carbon electricity sources (primarily additional hydro and geo-thermal, as discussed below) are deployed to compensate for them. However, in the scenario which the pathway adjusts from 2025, limited growth of solar and wind result in additional gas generation (and some coal generation) in the period to 2030, leading to higher emissions over this period, and subsequently greater reliance on net negative emissions later in the century, with a higher likelihood of overshooting the 1.5 °C target, as discussed below.

Fourth, in an "all adverse information" scenario, where DAC deployment is limited along with CCS and solar and wind deployment, and the carbon budget is reduced to 400 $GtCO_2$, the 2025 and 2030 respective emissions pathway adjustments are greatest. These scenarios also have the least net negative emissions in 2100 (3.0 $GtCO_2$/yr

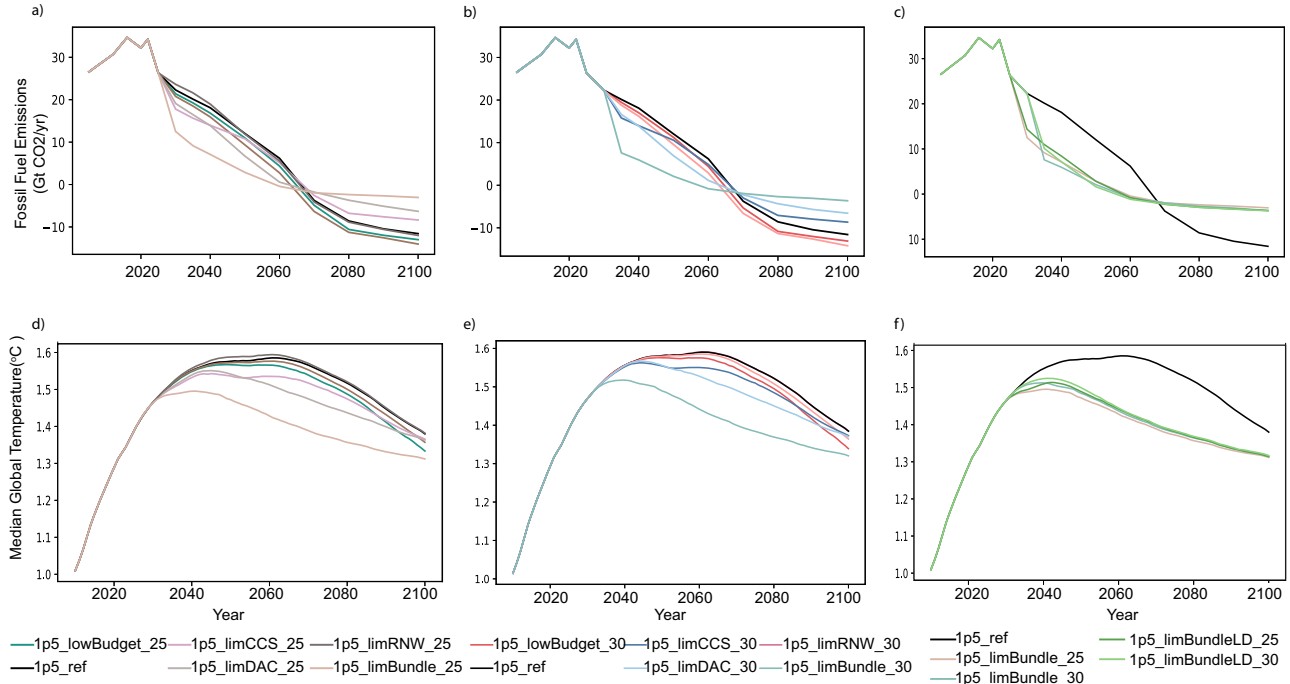

**Fig. 2 | $CO_2$ emissions from fossil fuel and industrial processes and temperature pathways.** Panel (**a**) shows $CO_2$ emissions in scenarios where new information arrives in 2023 and is acted on from 2025 onwards. Panel (**b**) shows $CO_2$ emissions in scenarios where new information arrives in 2028 and is acted on from 2030 onwards. Panel (**c**) compares $CO_2$ emissions in scenarios where all pieces of adverse information arrive in 2023 and 2028 respectively, plus additional scenarios where energy demand is assumed to be lower in light of this new adverse information. Panels (**d**)–(**f**) show the corresponding median temperature pathways (in °C) for each group of scenarios. On panels (**b**) and (**e**), the 1p5_limRNW_30 scenario line is hidden behind the 1p5_ref line. All scenario names are explained in Table 1.

with the 2025 pathway adjustment, and 3.6 $GtCO_2$/yr with the 2030 pathway adjustment). By necessity, to meet the stricter carbon budget without the possibility of so much $CO_2$ removal later in the century, these "all adverse information together" scenarios reach net-zero $CO_2$ emissions about a decade earlier than the 1.5 °C reference scenario.

The modelled pace of decarbonisation when all new adverse information is revealed and acted upon is much faster than in the 1.5 °C reference scenario. As shown most clearly by Fig. 2c), when adverse information is acted upon from 2025, this leads to a reduction in global emissions from 26 to 13 $GtCO_2$ in just five years, from 2025 to 2030 (2.8 $GtCO_2$ per year on average), compared to the 26 to 22 $GtCO_2$ reduction (0.8 $GtCO_2$ per year) in the 1.5 °C reference scenario in which no adverse information arrives. When adverse information is acted upon from 2030, the 22 $GtCO_2$ of emissions in that year more than halve to 8 $GtCO_2$ by 2035 (i.e. 2.9 $GtCO_2$ per year on average), compared to falling at 0.4 $GtCO_2$ per year in the 1.5 °C reference scenario. The fastest annual $CO_2$ emissions reduction in recent years, following Covid-19 lockdowns, occurred in 2020, with a reduction of ~2 $GtCO_2$[31]. Whether sustained reductions of this order are possible in the future cannot be firmly answered, although analysis of technological and economic dynamics, as presented in the following sections, can contribute to a consideration of this important issue.

The majority of the scenarios see an end of century temperature outcome of around 1.4 °C (Fig. 2d, e), with peak temperature between 1.5 °C and 1.6 °C (Fig. 3), placing them in the IPCC's "C1" category of "no or low overshoot" 1.5 °C scenarios[23]. The only exception is the "1.5_limRNW_25" scenario, which peaks just above 1.6 °C, owing to the substitution of fossil fuel power generation for renewables in the years following receipt of information that renewables will not scale as fast as initially expected (as discussed below).

Those scenarios in which all adverse information leads to a pathway adjustment (the "limBundle" scenarios) achieve lower peak temperatures than other scenarios. These require drastic near-term emissions reductions (as shown in Fig. 2a–c), which, using our non-$CO_2$ gas infilling assumptions (see Methods), are associated with a greater reduction in non-$CO_2$ emissions in the period to 2050, and hence a lower overall contribution to warming from these emissions. The real-world corollary to this purely non-$CO_2$ infilling model-driven result is that faster near-term action on decarbonising the energy and industrial process $CO_2$ emissions sectors could reasonably be expected to go in concert with more rapid decarbonisation of non-$CO_2$ emitting sectors. This is because some non-$CO_2$ emissions, such as methane, are heavily tied to fossil fuel usage and could be rapidly abated with a shift away from fossil fuels[32], as well as the fact that there is considerable potential to reduce non-$CO_2$ emissions at even relatively low carbon prices[33].

## Role of CCS and CDR
BECCS and DAC (both providing carbon dioxide removal, CDR), as well as fossil fuel CCS (in the power and—more notably—industrial sectors) are deployed in each scenario to different extents, relative to the 1.5 °C reference scenario in which no new adverse information arrives (Fig. 4). Total $CO_2$ sequestration is ~0.1 $GtCO_2$ by 2030 in all scenarios, so this time point is not shown in Fig. 4. For both the 2025 and 2030 pathway adjustment scenarios, a reduction in the 1.5 °C-consistent carbon budget, the shift of AFOLU from net sink to source, and limitations to DAC and solar and wind deployment lead to a greater deployment of BECCS, compared to the 1.5 °C reference scenario. By contrast, where there are limitations to CCS deployment, BECCS is much less prominent than in the 1.5 °C reference scenario. This is also the case for industrial CCS. There is relatively little fossil fuel with CCS deployment (in the power sector) throughout these scenarios owing to its relative lack of cost competitiveness and residual emissions.

Compared to the 1.5 °C reference scenario, DAC is deployed earlier when there is adverse information (other than the cases where DAC's own growth is limited). DAC in general only plays a minor role in

these scenarios to 2050, as in the 1.5 °C reference scenario it is already scaling at its maximum allowed rate of 12.5% annual installed capacity growth, leading to 0.1 $GtCO_2$ removal by 2040 and 0.5 $GtCO_2$ removal by 2050. Total CDR from DAC and BECCS does not exceed 4 $GtCO_2$ by 2050 in any of these scenarios, placing our analysis towards the more conservative end of assumed CDR availability in the first half of the century[13]. By 2100, this grows to over 17 $GtCO_2$ in the scenarios in which AFOLU cumulative emissions removals become positive cumulative emissions, contributing to net negative emissions of just over 14 $GtCO_2$ in these scenarios, as shown in Fig. 2.

The low energy demand scenarios lead to an overall reduction in reliance on CCS and CDR options by 2040, compared to the "all adverse information" scenarios without low energy demand.

## Energy system changes
When the carbon budget is decreased from 500 to 400 $GtCO_2$, the share of electricity in final energy serving the buildings, transport and industry sectors increases, to 33% in 2050 (for both the 2025 and 2030 pathway adjustment scenarios) compared to 30% in the reference 1.5 °C scenario (Fig. 5). A similar increase in electrification occurs when AFOLU net cumulative removals turn to net cumulative emissions. Electrification is much more marked in the other pathway adjustment scenarios, particularly those including limits to growth in CCS, where the share of electricity increases to over 40% by 2050. One exception is when renewables are limited, however, reflecting their role in making electricity increasingly cost-competitive with other forms of final energy. This means that other, more expensive forms of electricity step in, without increasing the electrification share above the reference 1.5 °C level. When all adverse information arrives at once, electricity increases to around 45% of final energy by 2050 in both the 2025 and 2030 pathway adjustment scenarios. Whilst this is a significant increase compared to the 1.5 °C reference scenario, it is still well within the range of IAM scenarios in the IPCC's sixth assessment report scenario database[34] (Supplementary Fig 3).

As well as changes to the rate of electrification of final energy, the pathway adjustments result in changes in the mix of electricity generation (Fig. 6). All adverse information scenarios with pathway adjustments from 2025 see greater hydropower generation in 2030 compared to the 1.5 °C reference scenario, whilst both the 2025 and 2030 pathway adjustment scenarios see greater hydro in 2040. This is more marked when DAC and CCS growth rates are limited, and in these scenarios there is additionally greater nuclear and solar generation, as well as some ocean-based power generation.

In the scenarios where the pathway adjustment begins in 2025, where wind and solar growth rates are limited, gas fills the generation gap by 2030 and there is also some additional coal generation, compared to the 1.5 °C reference scenario. There is ultimately a price to pay for these near-term fossil fuel emissions in terms of marginally greater net negative emissions towards the end of the century (Fig. 2a) and higher peak emissions and a greater likelihood of exceeding 1.5 °C (Fig. 3). Where the pathway adjustment begins in 2030, however, by 2040 there is no room for coal by 2040, in contrast to the 2025 pathway adjustment scenario in which some coal does remain by 2040.

In the all adverse information scenarios, there is greater electricity generation overall, reflecting the increased importance of electricity in final energy demand in these scenarios (as shown in Fig. 5). This also explains why, even with low energy demand, all adverse information scenarios ("1p5_LimBundleLD_25" and "1p5_LimBundleLD_30") have similar or higher electricity generation to the 1.5 °C reference scenario.

These scenarios have the greatest role for hydro and nuclear power, together accounting for 37% of global electricity generation by 2040, where the pathway adjusts from 2025 (and 33% when it adjusts from 2030), compared to 31% in the 1.5 °C reference scenario. Geothermal power is also notable, rising to 17% of total global generation

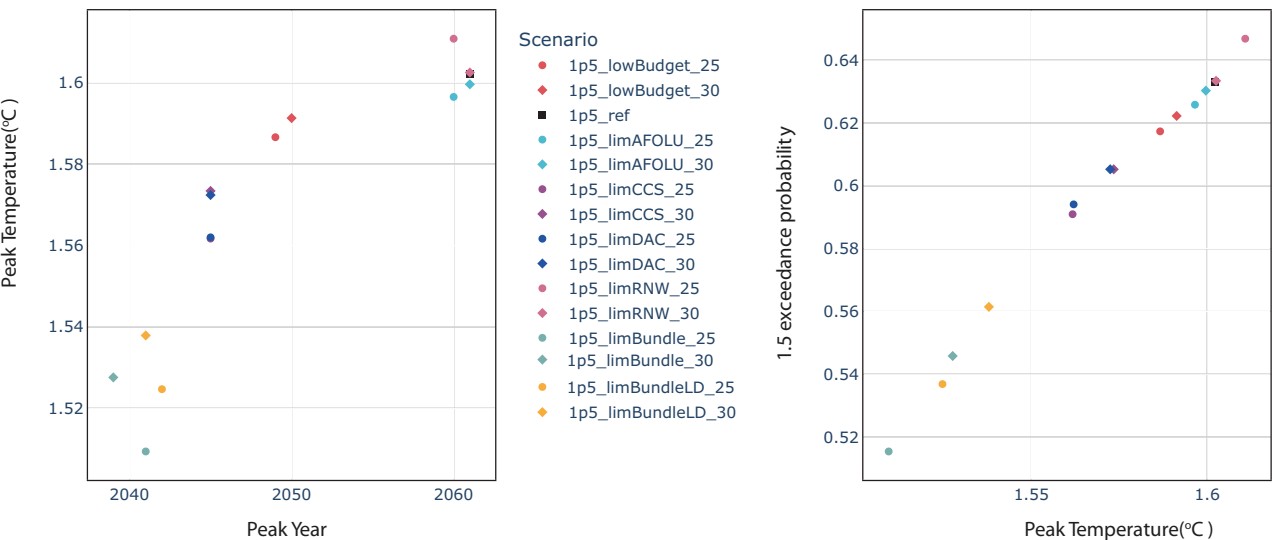

**Fig. 3 | Peak temperature outcomes of scenarios.** Left-hand panel shows peak temperature in °C plotted against peak temperature year. Right-hand panel shows probability of exceeding 1.5 °C, on the basis of probabilistic temperature calculated using the FaIR simple climate model. Black square marker denotes 1p5_ref scenario, circles represent scenarios in which the mitigation pathway adjusts from 2025, and diamonds represent scenarios in which the mitigation pathway adjusts from 2030. All scenario names explained in Table 1.

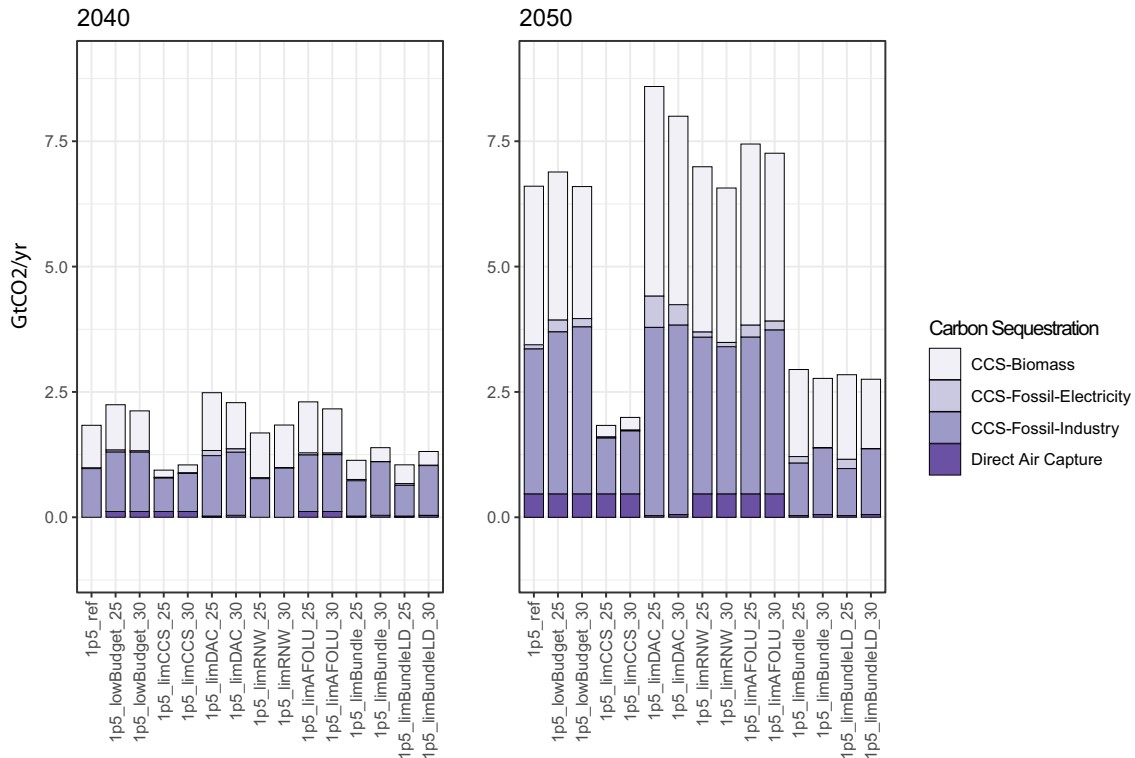

**Fig. 4 | Bioenergy and fossil CCS and DAC sequestration of CO₂.** Left panel shows 2040 sequestration of CO₂, and right panel 2050 sequestration of CO₂, in GtCO₂. CCS carbon capture and storage; DAC direct air capture. All scenario names explained in Table 1.

by 2040 in the 2025 pathway adjustment scenario with all adverse information, compared to 5% in the 1.5 °C reference scenario. This geothermal generation would require of the order 0.5–1.0 TW of global capacity, compared to an approximate 5 TW global capacity that could be built at a cost of €50/MWh or lower[35] (though whether it could be scaled that fast, and in a sustainable way, remains unclear).

Ocean power also appears as a further renewable electricity option as early as 2030. The total generation from ocean power,

at 10 EJ/year by 2040 in the "all adverse information" 2025 pathway adjustment scenario ("1p5_LimBundle_25"), is highly ambitious. Although ocean energy potential is vast (estimated at over 250 EJ/year[36]), the International Renewable Energy Agency (IRENA)'s 1.5 °C "World Energy Transitions Outlook" scenario has just 350 GW of global installed capacity by 2050[36], equivalent to about 6 EJ/ year of generation at an assumed 50% capacity factor. Thus, our scenarios would imply a considerably faster rate of realisation of the global potential.

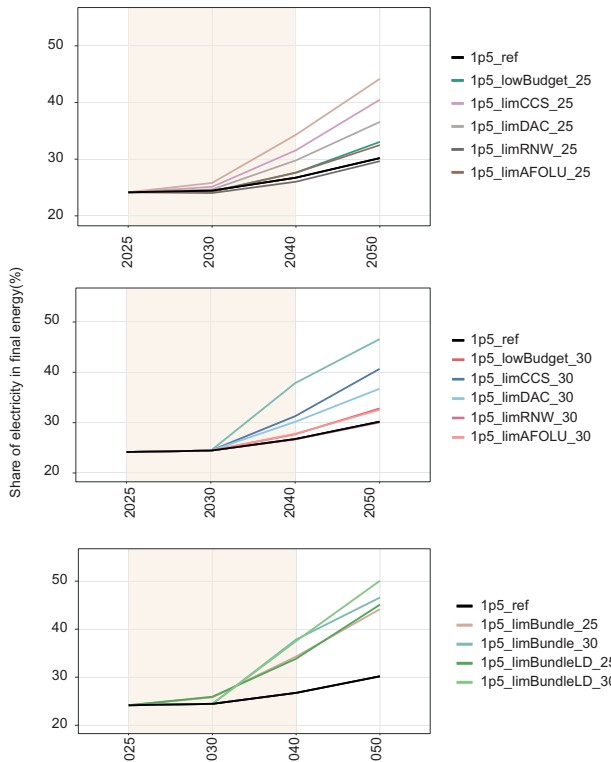

**Fig. 5 | Electrification of final energy.** Panels group scenarios as those with pathways adjusting from (upper panel) 2025, (middle panel) 2030 and (lower panel) comparing "all adverse information" scenarios with and without low energy demand for the 2025 and 2030 pathway adjustments. Limited renewables scenarios are hidden behind the 1.5 °C reference scenario in middle panel, indicating no increase in electrification in these scenarios, compared to the 1.5 °C reference scenario. All scenario names explained in Table 1.

The low energy demand scenarios see lower overall electricity generation, as expected, and a consequently lower absolute level of reliance on geothermal power.

As well as increased electrification of final energy, as shown in Fig. 5, adverse information also results in reduced final energy demand (primarily driven by the price elastic response to increased energy prices), compared to the 1.5 °C reference scenario. For example, where the pathway adjustment begins in 2025 in response to all adverse information, transport final energy reduces to 10% below the 1.5 °C reference scenario level by 2030, and by 22% by 2040 (Fig. 7). For industry, the equivalent figures are 18% by 2030 and 9% by 2040, and for buildings, 15% by 2030 and 17% by 2040. The sectoral final energy reductions by 2040 (compared to the reference 1.5 °C case) in the 2030 pathway adjustment scenario are approximately equal to those the 2025 pathway adjustment scenario, implying a faster rate of reduction over the period 2030–2040, since the adjustment begins five years later. Furthermore, the low demand scenarios, as expected, lead to a deeper reduction in energy end-use demand across the transport and industry end-use sectors by 2040. For example, in the 2025 pathway adjustment scenario with all adverse information and low energy demand, transport final energy reduces by 26% below the 1.5 °C reference scenario level by 2040 (compared to 22% without low energy demand), industry final energy by 20% (versus 9% without low energy demand), and buildings final energy 21% (versus 17% without low energy demand).

## Costs

The carbon price in the 1.5 °C reference scenario is (2010US)$86/tCO$_2$ in 2030, and $114/tCO$_2$ in 2040 (Fig. 8). This rises in all scenarios where

adverse information is received, and most markedly, where there is limited growth in CCS and DAC – each leads to a carbon price of $200/tCO$_2$ or more by 2030, in the 2025 pathway adjustment scenario. Limits to solar and wind growth do not have a discernible impact on the carbon price profile, indicating that these and their replacement technological solutions are not the marginal measures to achieve the decarbonisation pathway. In the "all adverse information" scenarios, the carbon price rises very steeply from 2025 or 2030, depending on when the pathway is adjusted. For example, in the 2025 pathway adjustment scenario, the carbon price increases to $1362 /tCO$_2$ by 2030, a ten-fold increase on the 1.5 °C reference scenario. Even in the low energy demand variant of this scenario, the carbon price is $531 /tCO$_2$ by 2030. The story is similar to the 2030 pathway adjustments, which see a ten-fold increase in carbon price by 2035, even in the low energy demand variant. There is a clear benefit to earlier action, with much lower carbon prices – indeed a hypothetical scenario in which all adverse information is assumed from the outset (i.e. from the start of 2023, results in a carbon price of $850/tCO$_2$ by 2030—far below the $1362 /tCO$_2$ reported above for the 2025 pathway adjustment scenario (Supplementary Fig 4).

Mitigation costs (as expressed as a share of GDP in a scenario in which only current policies are simulated) reflect these carbon price variations. Where the pathway adjustment occurs in 2025, the mitigation cost is 0.25–0.45% of GDP by 2030 for each scenario in which a single adverse piece of information is received, compared to 0.24% of GDP in the 1.5 °C reference scenario (Supplementary Fig 2). When all adverse information is received in 2025, the mitigation cost is 1.62% of GDP in 2030. When the pathway adjustment occurs in 2030, then each individual adverse piece of information leads to a mitigation cost of 0.47–0.82% of GDP by 2040, compared to 0.47% of GDP in the 1.5 °C reference scenario. When all adverse information is received together and the pathway is adjusted from 2025, the cost is 3.9% of GDP by 2050—around four times the cost of the 1.5 °C reference scenario. With a pathway adjustment from 2030, by 2050 the mitigation cost is 5.0% of GDP. Clearly, this is a considerable cost increase compared to the 1.5 °C reference scenario, but it is important to place it in the context that it doesn't account for any co-benefits to climate action, any policy mechanisms that might resolve distortions in the economy, nor (most importantly) the costs deriving from climate impacts associated with less aggressive mitigation[37].

Nevertheless, one thought experiment is to consider what warming would result if all adverse information were received in 2028 and then acted on by 2030, but with a mitigation expenditure that matches that of the 1.5 °C reference scenario. We simulate such an experiment, finding that the minimum achievable 2020–2100 carbon budget whilst keeping the mitigation cost to 2050 approximately the same as for the 1.5 °C reference scenario is 1100 GtCO$_2$, leading to a peak warming of 1.64 °C in 2089 (Supplementary Fig 2).

## Summary indicators

A comparison across major indicators reveals that where expectations are updated in light of adverse information, this tends to lead to higher rates of emissions reductions, final energy demand reductions, rates of electrification increase and higher costs in the decade to 2040 (Fig. 9). Adverse information also leads to increased reliance on CDR (BECCS and DAC) in the latter half of the 21st century, where these are not themselves limited. Where all adverse information is received at once, this leads to much greater rates of near-term decarbonisation (utilising low-carbon electricity sources such as geothermal, hydro and nuclear) and less reliance on CDR throughout the 21st century. Consequently, these scenarios show a lower probability of overshooting the 1.5 °C temperature limit. However, the carbon prices associated with such scenarios, even where energy demand is assumed to be lower, are very high, at over $1000/tCO$_2$ by 2040.

## Discussion

Adjusting pathways to keep a 1.5 °C temperature limit is hypothetically possible with the technological options assumed in the integrated assessment model utilised here, even when multiple sources of information are received simultaneously, as late as the end of the second global stocktake in 2028, when acted upon from 2030. However, this

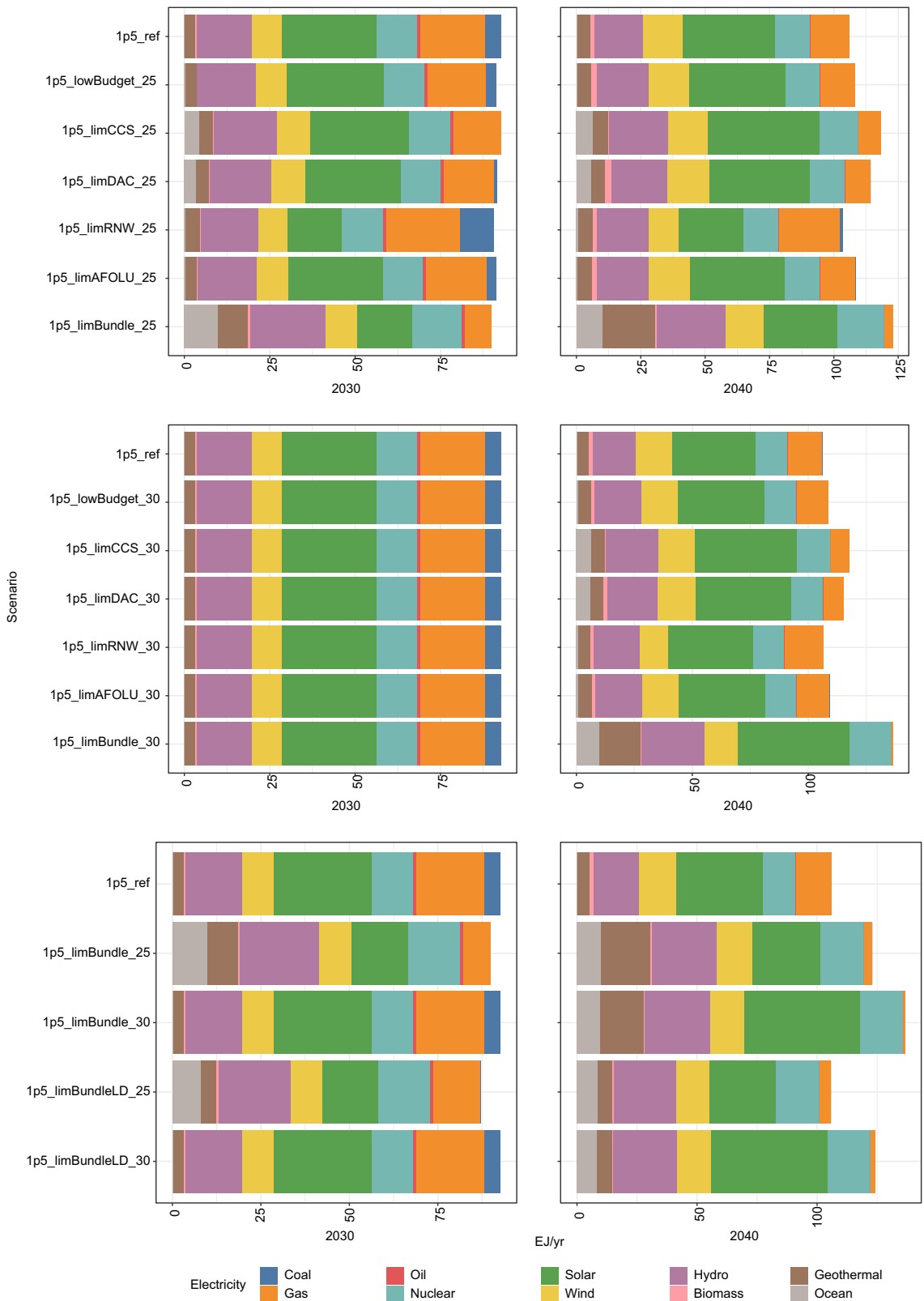

**Fig. 6 | Electricity generation over period 2030–2040, by technology.** Panels group scenarios as those adjusting their pathway from 2025 (top), 2030 (middle) and comparing "all adverse information" scenarios with and without low energy demand for the 2025 and 2030 pathway adjustments (bottom). Scenarios in which the pathway adjusts from 2030 onwards have identical 2030 values, by design. All scenario names explained in Table 1.

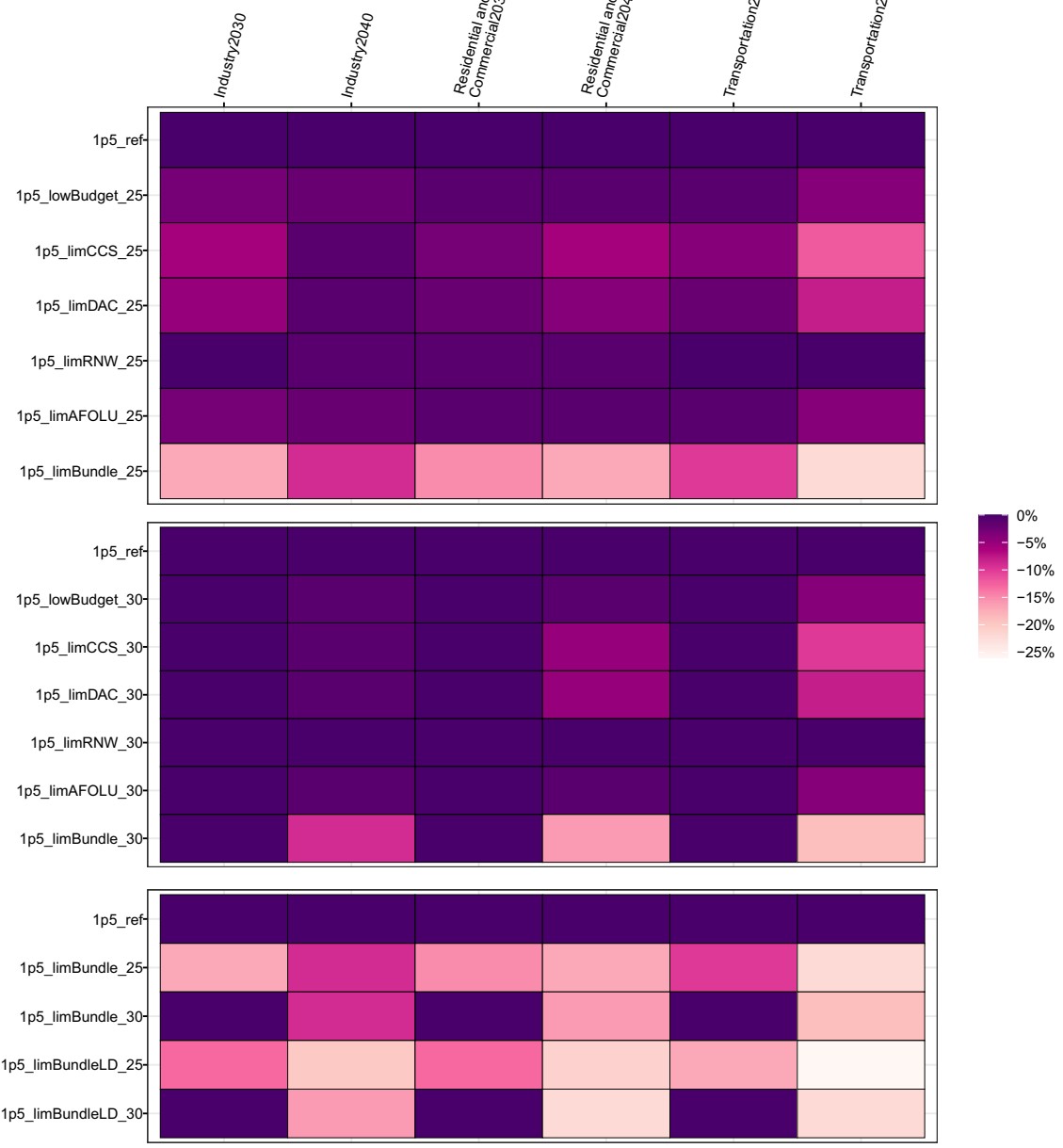

**Fig. 7 | Final energy demand in transport, industry and buildings in 2030 and 2040.** Panels group scenarios as those adjusting their pathway from 2025 (top), 2030 (middle) and comparing all adverse information scenarios ("limBundle") with and without low energy demand for the 2025 and 2030 pathway adjustments (bottom). All values for final energy per sector are normalised to the 1.5 °C reference scenario ("1p5_ref") which has a value of 1. All scenario names explained in Table 1.

requires drastic increases in the rate of emissions reductions, sustained at almost 3 GtCO₂ per year until 2035, which is seven times the rate of the 1.5 °C reference scenario in which no adverse information is received. As a result, annual mitigation costs by 2050 are some five times as high as in the 1.5 °C reference scenario, and the carbon price is almost \$4000/tCO₂ by 2040.

It is less costly to adjust the mitigation pathway if this adjustment begins from 2025, rather than from 2030. This reaffirms the messages from the literature on "cost of delay" of mitigation action. For example, one previous study shows that by 2050, a 2 °C-targeted mitigation pathway loses approximately 4% of global GDP (relative to an NDC-only reference case) when action starts in 2020, and 5.6% when action starts in 2030 (i.e. a 40% increase in mitigation cost by this time as a result of delay)[6]. Another study shows an 18% higher cumulative mitigation cost over the period 2010–2100 where mitigation is delayed

from 2020 to 2030[38]. By comparison, our study sees 2050 mitigation costs around 25% higher in the "all adverse information" 2030 pathway adjustment scenario, compared to the 2025 pathway adjustment scenario. This finding thus highlights the similar relative magnitude of delayed mitigation pathway adjustments, when compared to the additional cost of commencing any mitigation later.

Where key technologies are not available, there are others that could be available to step up. For example, when CCS and DAC are limited, there is an increase in renewables (principally hydro and solar) and nuclear generation, and greater electrification of final energy. Where renewables growth is limited, then—given that wind and solar are amongst the cheapest electrification options, this drives some additional demand for fossil power in the near term, particularly when the pathway adjustment begins in 2025. In this case, there is greater demand for CDR longer-term, with higher peak temperatures and a

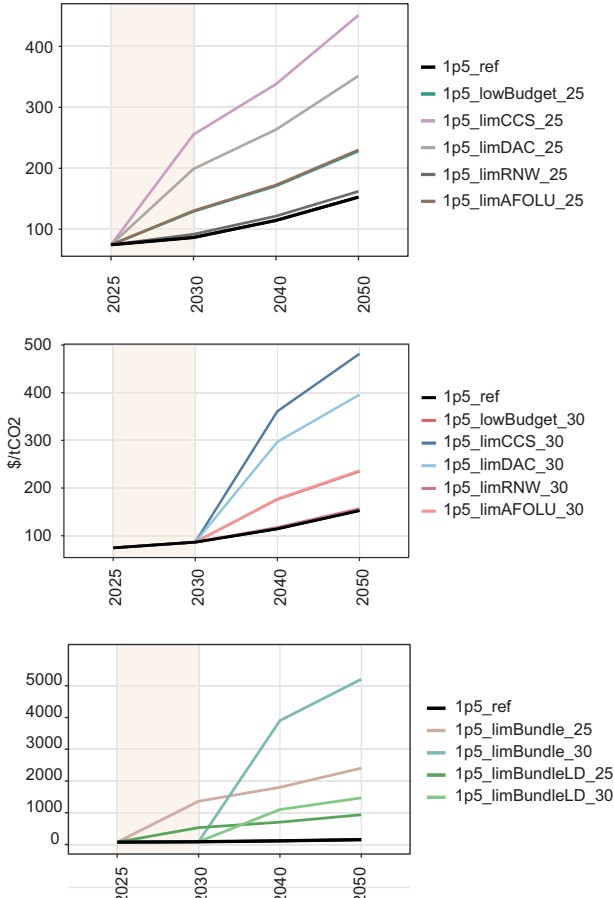

**Fig. 8 | Carbon price in scenarios.** Panels group scenarios as those adjusting their pathway from 2025 (top), and from 2030 (middle) with bottom panel comparing "all adverse information" scenarios with low energy demand variants. In the top panel, the "1p5_lowBudget_25" and "1p5_limAFOLU_25" lines overlap. In the middle panel, the "1p5_ref" and "1p5_limRNW_30" lines, and the "1p5_low-Budget_30" and "1p5_limAFOLU_30" lines overlap. All scenario names are explained in Table 1.

greater likelihood of exceeding 1.5 °C as a result. Energy demand reductions can help the economic feasibility of achieving the 1.5 °C limit, leading to lower carbon prices when all adverse information arrives together, compared to standard energy demand scenarios. Nevertheless, even with low energy demand, the carbon price by 2040 is still $700/tCO$_2$ when the pathway adjusts from 2025, and $1100/tCO$_2$ when the adjustment begins in 2030.

It should be noted that the analysis here reflects only one model result, and others may find differing outcomes, including around the relative role of nuclear, hydro and geothermal, as well as fossil with CCS technologies, depending on their underlying model structure, as well as technology cost, deployment rate and substitutability assumptions. The TIAM-Grantham model overall exhibits a "medium" response to enforced mitigation objectives, considering key diagnostic indicators including emissions reductions relative to a no-policy baseline; the balance between supply and demand side measures in decarbonisation; and rate of fossil fuel reductions[39]. Its 1.5 °C reference scenario shows metrics in general within the range of other IAM models for 1.5 °C scenarios (Supplementary Fig 8), indicating it is not an outlier, but nevertheless, technology-specific outcomes from this study should be treated with caution.

Nevertheless, some clear policy implications can be drawn from this analysis. Most obviously, it is important to not peg hopes solely on one, or even a few, mitigation strategies, but to continue developing all options towards commercial readiness, including the skills and

infrastructure to achieve rapid scale-up if needed. A broader portfolio approach such as this may be more expensive in terms of earlier stage R&D investment, demonstration and deployment support, compared to a strategy that targets the majority of investment towards what may initially appear to be least-cost options. Specifically, our analysis highlights that if CCS and DAC fail to deliver as initially planned, then additional investment/planning around solar and wind, hydropower, energy demand reductions and nuclear are advisable. Energy demand reduction also enhances the feasibility of meeting the 1.5 °C goal if renewables growth stalls—as does CDR, to draw down emissions as soon as possible, given the possibility of near-term overshoot resulting from fossil generation filling in the renewables gap. Our analysis also highlights—perhaps unsurprisingly—how much of a knife-edge the 1.5 °C goal sits on. By 2040, the mitigation cost is around 3–4% of GDP if there are growth and deployment limits to all of renewables, CCS and DAC. This compares to less than 1% of GDP if only single technology deployments are limited, and gives a sense of the value (i.e. 2–3% of global GDP by 2040) of ensuring "back-up" mitigation options (including, importantly, energy demand reduction options) should one technology or group of technologies fail.

Further analysis should investigate in detail the costs and policy implications of such earlier investments in potential resilience measures. Hedging strategies around not only technology failure or under-performance (in terms of deployment growth rates) but also around multiple risks and vulnerabilities, could be identified by analysing a very large ensemble of different mitigation pathways which are hypothetically subjected to multiple risk realisations. This would help identify those least vulnerable to risks, as well as actions to minimise risks, as part of a robust decision-making approach[40,41]. Critically, as already highlighted through multiple other analyses and confirmed here, energy demand reduction through efficiency measures and demand reduction behaviours is a central strategy towards lower-cost mitigation pathways. Nevertheless, this study starkly highlights that there is now little room for manoeuvre—or failure—if we are to meet the 1.5 °C target. Given the conclusion of the first Global Stocktake this year (i.e. in 2023) the first mitigation pathway adjustment scenario (which sees a course adjustment in 2025) could reasonably be taken to represent actions and understanding around adverse information that begins in the very near future.

We only explore single-time pathway adjustments in 2025 and 2030, whereas in reality, there is likely to be a need to more frequently adjust mitigation pathways in light of continually emerging evidence. Such evidence will not just include the conclusive arrival of adverse information, but rather revisions to investment prospects of different technologies, as innovation, policy and societal attitudes to technologies evolve. For example, the 2010s have widely been seen as a "lost decade" for CCS[42], yet rather than this resulting in a firm view that CCS will have only a limited role to play, it is by contrast attracting new investment and policy support across different countries, resulting in record levels of growth[43]. As such, our stylised representation of the arrival of adverse information, with no future prospect of upward revision, may be deemed conservative or precautionary in nature.

Further analysis could include an explicit examination of scenarios which are ex ante designed to meet 1.5 °C-consistent carbon budgets without the need for CO$_2$ removal after exceeding the budget (i.e. "net zero" budgets). Alternatively, further analysis could also explore how net zero worlds would differ depending on positive technology surprises and availability, such as the emergence of nuclear fusion, or more drastic behavioural and societal changes around energy and material production and consumption. In addition, the land use implications of mitigation scenarios, including a greater focus on agriculture, forestry and land use (AFOLU) and its interplay with food prices and biomass availability, are important to explore where AFOLU provides significant net emissions removals.

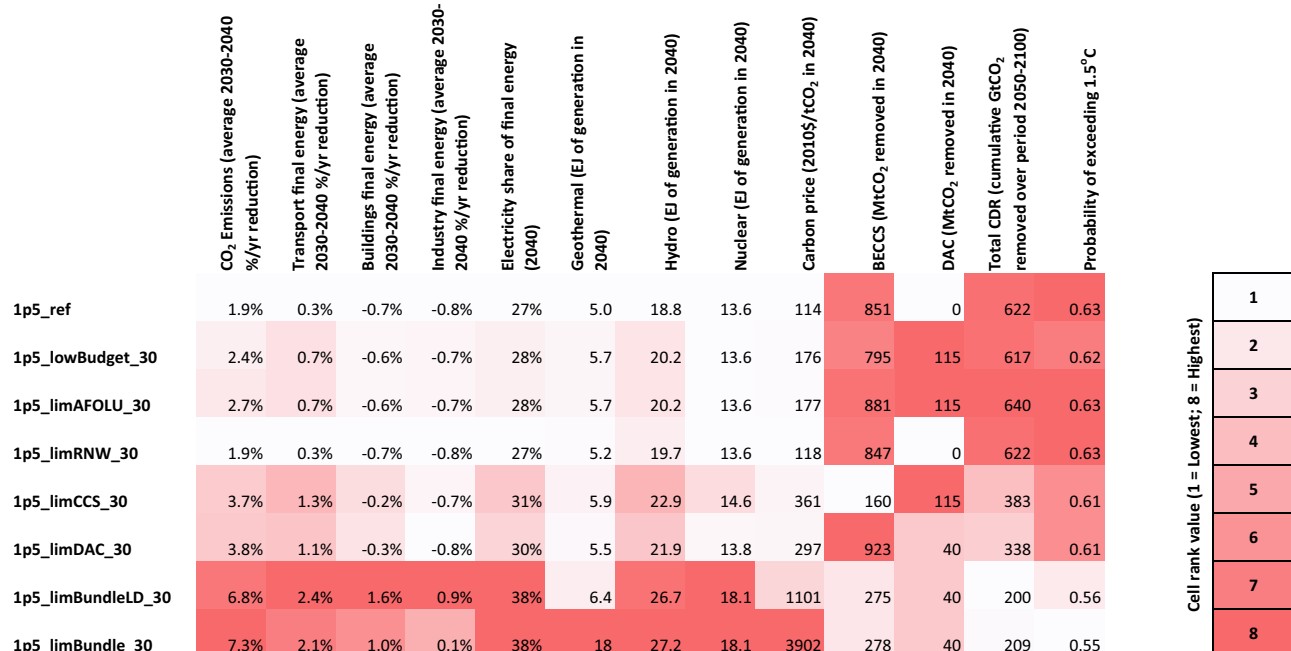

**Fig. 9 | Summary indicators across 2030 pathway adjustment scenarios.** Indicators are rank coloured, with white referring to scenario with lowest value and red referring to scenario with highest value, for each indicator. BECCS bioenergy with carbon capture and storage, DAC direct air capture, CDR carbon dioxide removal. All scenario names are explained in Table 1.

Overall, however, we demonstrate the reduced likelihood, as well as reduced economic feasibility, of remaining below a 1.5 °C target in light of adverse information, particularly with multiple simultaneous adverse occurrences. Our results can be generalised for policy and decision makers to highlight the economic and technological fragility of achieving the 1.5 °C target, particularly in light of new adverse information, the consequent value and benefits of investment in a range of low-carbon options rather than a select few, as well as the benefits of lower energy demand. On the latter point, more profound changes in behaviour or economic system operations (not represented in our framework) could provide further risk-mitigation options. With a 1.5 °C future now on a knife edge, robust planning strategies that consider all possible options are now more essential than ever.

## Methods
### Modelling tools
The modelling approach is shown schematically in Supplementary Fig 1. Our integrated assessment model (IAM), TIAM-Grantham (see online documentation[44] and code availability[45]), representing fossil fuel and industry $CO_2$ emissions, is set up with resolution to represent the periods 2020, 2022, 2025, and 2030, and then ten-year timesteps from 2030 onwards. TIAM-Grantham operates as a perfect foresight cost-optimisation model, which means it calculates a least-cost mitigation pathway to meet a prescribed cumulative $CO_2$ emissions level with full knowledge of future costs and availability of all energy and industrial technologies and fuels. The model can be run to any given time step (in this case 2025 and 2030) and then re-started from that time step with new information assumptions inputted into it. In this way, it is used to simulate the pathway adjustments explored in this study. We use SSP2 assumptions on population and economic growth, adjusted to reflect outturn data up to 2020[46]. The scenario design also requires assumptions on agriculture, forestry and land use (AFOLU), which we derive from both existing IPCC scenario databases and our own use of the MAgPIE land use model (version 4.4)[28], as detailed below.

In order to determine the temperature effects of these emissions pathways, we need estimates of non-$CO_2$ emissions. We obtain these by

harmonising a complete dataset of emissions, inferring the levels of other emissions by infilling, and finally running this through a reduced complexity climate model. First, we obtain a comparable set of emissions data. This is done using scenarios from the IPCC AR6 database[34]. We harmonise the $CO_2$ emissions in these data to match our $CO_2$ paths by a constant offset. Other variables are harmonised to historic data obtained from the CMIP6 emissions database, based on refs. 47–50, using a multiplicative factor that starts at the value required to unify the data in 2015 to the historic value and tapers to 1 in 2050. Values before 2015 are set equal to the historic value. For emissions where no historic data is available from these sources (i.e. the F-gases), no harmonisation is enacted.

Secondly, we use the module Silicone[29] to establish relationships between total $CO_2$ emissions and the other emissions required to run a simple climate model and infill results based on this. For non-F-gas emissions we can infill this using only the SSP2 scenarios in the SR1.5 database. We use the quantile regression technique "quantile rolling windows" to find the median level of each species emitted given the level of total $CO_2$ emissions in a given year. For F-gases, none of these scenarios have a complete set of emissions, so we calculate the total F-gas emissions using this technique, then break down the F-gas total into SF6, PFCs and HFCs, which are broken down into their components in turn, using any scenarios in the database with the required set of emissions. This takes place using the "decompose collection with time-dependent ratio" function. The MAGICC default set of historic emissions is appended for all species prior to 2005, which is based on the SR1.5 REMIND-MAgPIE set of emissions.

This set of data is then run through a climate model emulator, FaIR version 1.6.4[30]. This has a simple model of the climate with a range of variables that are constrained to match both historic warming since 1765 and the IPCC's assessment of key climate values, documented in IPCC WG1[22] (Chapter 7 cross-working group box 1). We project temperature trends with probabilities represented by the fraction of runs reaching a given temperature.

### Scenario design
Our scenario design is centred on exploring questions around how the world as a whole should adjust an initially-planned cost-optimal

mitigation pathway in light of the arrival of new information. We simulate the reception of this information at two time-points: the first point is at the conclusion of the first global stocktake (GST), at the end of 2023. We assume the world then has a year to respond to this new information and make necessary adjustments to its mitigation pathway, from the beginning of 2025. A second-time point for the reception of new information is the conclusion of the second GST, at the end of 2028, leading to an adjustment of the mitigation pathway in 2030. We do not explore mitigation pathway changes in both 2025 and 2030 in the same scenario, though this would be an obvious extension of the analysis. Rather, our purpose is to highlight the extent of whether, and if so how, mitigation pathways can be changed to keep the 1.5 °C target feasible, both when pathways are changed as early as 2025, or as late as 2030. This scenario design is therefore deliberately intended to test the potential benefits (in terms of the feasibility of the 1.5 °C target) of receiving information earlier in this decade, which seems intuitively more likely to lead to a mitigation pathway adjustment that keeps the 1.5 °C target alive, compared to receiving information late in the decade. We formalise our scenario design through the following specific research questions:

If global society embarks on a cost-optimal mitigation pathway from now, then how do we adjust the pathway to achieve the 1.5 °C target, if new (adverse) information arrives in 2023?

As above, but considering the arrival of new information in 2028, and a ramping up of technologies and measures in 2030.

Of course, more frequent pathway adjustments could also be simulated (for example, adjustments in both 2025 and 2030, or at intervening times), to further explore the implications of more frequent updates to pathways. We keep our analysis circumscribed to these two illustrative time points for brevity. Within this high-level scenario framework, we make several specific assumptions to increase the realism of near-term mitigation pathways. First, global coordinated mitigation is assumed to begin from 2023 onwards in an initial cost-optimal scenario which assumes no adverse information. Until that time, major current policies (as of 2021) are followed in each region, with specific details of policies considered in ref. 46 This reflects the likelihood that emissions will rise through 2022, as they did in 2021, following recovery from the covid pandemic[1]. In addition, we do not allow our scenarios' energy sector $CO_2$ emissions levels to fall below 27.5GtCO$_2$ in 2025, reflecting the lack of realism in being able to rapidly shift away from high-carbon infrastructure in very short timescales, as well as inter-regional equity considerations[12].

Our initial, cost-optimal scenario's objective is to achieve a 1.5 °C-compliant carbon budget, assumed to be 500 GtCO$_2$ from the start of 2020, in line with IPCC AR6 WGI's 50% likelihood budget[22] (IPCC, 2021). The majority of our scenarios use a 3% discount rate to value future mitigation costs, as well as what we deem to be conservative assumptions around the potential for carbon dioxide removal via afforestation, bioenergy with carbon capture and storage (BECCS) and direct air carbon capture and storage (DAC), as detailed below. The intention of these scenario design features is to prevent any significant carbon budget overshoot, so as to concentrate the focus on pathway adjustments on the near-term mitigative actions necessary to meet the Paris Agreement goals, without simply assuming that any adverse information will lead to little near-term change, and instead a "mop-up" of excess emissions later in the century (Supplementary Fig 7).

We use Warszawski et al.'s[51] identity for a carbon budget to guide our attention on the types of new information that could be received in 2023 and 2028. Specifically, Warszawski et al.[51] show that the carbon budget consistent with a given temperature goal is the sum of cumulative $CO_2$ emissions from the energy sector (the result of the carbon intensity of energy multiplied by annual energy demand), $CO_2$ emissions from industrial processes such as calcination of limestone in cement manufacture, net $CO_2$ emissions from the agriculture, forestry

and land use (AFOLU) sector, fewer $CO_2$ removals from carbon removal technologies such as BECCS and DAC.

We focus our attention on the size of carbon budget consistent with the 1.5 °C target, the availability and scale-up rate of technologies (wind and solar, and CCS) to reduce the intensity of energy supply, the scale-up rate of CCS as a key decarbonisation technology for industrial process emissions as well as (with biomass) $CO_2$ removal, and the scale-up rate of DAC as a further $CO_2$ removal measure. We also use a sensitivity around AFOLU net emissions or removals, as well as around energy demand levels.

First, we assume the carbon budget consistent with 1.5 °C is decided to be 400 GtCO$_2$, rather than 500 GtCO$_2$, reflecting recent updates to the estimated carbon budget when comparing IPCC WGI[22] with WGIII[23] estimates. Next, we implement a drastic downward revision of CCS annual capacity growth, from an initial 20% per year, in line with the fastest historical emerging technology growth rates[52], to 3% per year, reflecting pessimism around CCS's prospects following its lack of deployment over past decades[53]. We also reduce the initially-envisaged scale-up rates of DAC, set at 12.5% per year in existing TIAM-Grantham modelling to achieve expert-derived "reasonable" levels of DAC growth[13], also to 3% per year, assuming the loss of investor confidence and/or lack of cost-reduction possibilities. Additional concerns around the deployment of CCS and DAC, including around $CO_2$ storage permanence (if storage monitoring regulation is poor[54]), or the lack of investment and preparatory actions to achieve high sustainable $CO_2$ injection rates[55], could be further feasible factors that slow down deployment rates.

For solar and wind, we reduce initial rates of 30% per year and 10% per year, respectively, as used in the current TIAM-Grantham modelling, with downward revisions to 6% and 5%, respectively, following analysis to suggest growth will be Gompertz-like (i.e. slow down over time), rather than continued exponential[24]. The final piece of adverse information concerns AFOLU, where we use an initial cumulative 136 GtCO$_2$ of net $CO_2$ cumulative removals over the period 2018–2100 (derived from the IPCC SR1.5 database median pathway for all 1.5 °C scenarios[56]), and shift this to net positive $CO_2$ emissions of 46 GtCO$_2$ over the period 2018–2100. The former figure is close to that for 1.5 °C scenarios in the "C1" and "C2" categories in the IPCC AR6 scenarios database (144 GtCO$_2$)[34]. The latter figure is derived from a simulation using the MAgPIE land use model (version 4.4)[28], with current policies and no carbon pricing, to reflect relatively poor land use policies. This compares with a median net cumulative emissions level over the period 2020–2100 of 21 GtCO$_2$ (range −377 to +255 GtCO$_2$) for the six modelled SSP2-4.5 scenarios (as derived from the SSP database of shared socio-economic pathways[57]), thereby representing a reasonably close match to an RCP pathway (i.e. RCP4.5) that could be deemed representative of current policies and targets[58].

In addition, our scenarios also consider the imposition of a remedial action via energy demand reduction, applied to scenarios where all adverse information is received together. The rationale is to explore what benefits lower demand might bring, in terms of reduced costs or increased feasibility of meeting the 1.5 °C target, in light of these multiple adverse sources of information. We base these low energy demand assumptions on analysis detailing a set of realistic possibilities to reduce demand through both technological and behavioural measures in the industry, transport and buildings sectors[59], with resulting demand level data in both the standard and low energy demand scenarios available in ref. 60.

We do not explore the full range of scenarios with several pathway adjustments, but rather a stylised set of scenarios with single pathway adjustment episodes. This, nevertheless, sheds light on a range of potential adjustment strategies, likely to be of relevance to real-world instances of more frequent adjustment requirements.

## Data availability

The results data generated in this study have been deposited in the Zenodo database with unrestricted access: https://doi.org/10.5281/zenodo.8118060.

## Code availability

Code underlying the TIAM-Grantham model is available at: https://github.com/etsap-TIMES. Code underlying the FaIR model is available at: https://github.com/OMS-NetZero/FAIR/tree/v1.6.4. Code underlying the Silicone model is available at: https://github.com/GranthamImperial/silicone/. Code underlying the MAgPIE model is available at: https://github.com/magpiemodel/magpie/tree/v4.4.0.

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

## Acknowledgements

This work received funding from the UK Department for Business, Energy and Industrial Strategy in 2021/2022. A.G., A.H. and S.M. acknowledge support from the Horizon Europe R&I programme project IAM COMPACT (Grant no. 101056306). A.H. also acknowledges support from the Climate Compatible Growth (CCG) Programme (IATI: GB-GOV-1-300125) of the UK's Foreign, Commonwealth & Development Office. The sole responsibility for the content of this paper lies with the authors: the content does not necessarily reflect the views or official policies of the UK Government or European Commission.

## Author contributions

A.G. and J.A.L. conceived of the study. N.G. reviewed previous relevant literature. A.G. and S.M. led the scenario design, with contributions from R.D.L., N.G., A.H., A.K., J.R. and J.A.L. S.M. ran the TIAM-Grantham model and prepared all of the results figures. A.K. ran the MAgPIE model. R.D.L. ran the Silicone model. D.B., R.D.L. and L.G. ran the FaIR model. A.G. wrote the manuscript, with contributions from S.M., R.D.L., N.G., D.B., L.G., A.H., A.K., J.R. and J.A.L.

## Competing interests

The authors declare no competing interests.
