## [Peer Review File · Nature Communications]

Adjusting 1.5 degrees C climate change mitigation pathways in light of adverse new informationREVIEWER COMMENTS

Reviewer #1 (Remarks to the Author):

The authors explore how global cost-optimal mitigation scenarios, complying with a 1.5C target, change upon the arrival of new adverse information in either 2023 or 2028 to achieve the same climate target. This work touches upon a critical blindspot of IAMs, i.e. the models that generate that kind of scenarios: the lack of adaptiveness and/or robustness of those scenarios to new information, which inevitably will impact their optimality and feasibility. Although the study has the potential to be relevant, significant limitations must be taken care of.

First, the term "course correcting" in the title and throughout the paper seems a bit of a stretch with respect to the extent of the analysis. The "correcting" is a 1-time response to a quite limited set of scenarios, either in 2025 or 2030, over the whole century. Figure 1 shows the classic dynamic adaptive pathways, but the original concept is much more dynamic. Furthermore, a paper [1] that tries to do so for a simple IAM, which I think is relevant, is not cited. Possible suggestions are to change the "course correcting" expression to something less ambitious, to explain the implications of not considering further correction points, and to discuss the acceptability of such simplification.

Second, this is a one-model study, and as such, it needs to address the problem of representativeness and neglected uncertainty. I recommend taking the extra effort to locate the TIAM-Grantham model within the landscape of available models, especially in terms of technological preferences and other diagnostics [2,3], for example with respect to what can be found in the last IPCC AR6 scenario database. Also note that other studies have looked at scenarios with delayed action and limited technologies in the past [4,5, for example]. A comparison with previous results and highlighting of the novelties seem to be missing here.

Third, among the "clear policy implications" listed at the end, the importance of a diverse technology portfolio is mentioned. Nonetheless, it is unclear how the analysis supports this claim. Qualitatively, this is trivial from the assumption that if adverse things can happen to any technology, a diverse portfolio becomes an insurance. The value added here would be the whole quantification exercise, which is again hindered by being a 1-model study, for which "technology-specific outcomes should be treated with caution". Make sure to garner all the evidence you have or need to prove that a mixed-tech scenario is indeed beneficial when considering all the potential course corrections, and possibly show what the costs are of not doing so.

Fourth, the adverse information is conclusive about the possibility of relying or not on a technology, or its extent. Is this a reasonable assumption? Please extract from the relevant cited literature the key points that make it credible to have such dichotomic scenarios. Also, why considering a DAC scenario even if it seems to play a relatively minor role, according to Fig 4? And why does the LimBundle LD have the same or more electricity generation than the ref, even if it is a low-demand scenario?

Fifth, the model provides results for any course correction combination, no matter the delay or the tech limit. But are these results feasible at all? Otherwise, this would provide a false sense of serenity that a solution always exists. Are the high numbers in the last row of figure 9 reasonable (e.g. compared to historical statistics and/or AR6DB)? Even the ocean becomes a significant energy source (Fig 6), which is not common in this literature.

Lastly, I would improve the readability of some graphs, e.g. in Fig 2 it is almost impossible to track which line is which scenario; in Fig 3 the points are very sparse and hard to distinguish; in Fig 8b only 4 lines are distinctly visible, yet 6 are in the legend. Furthermore, the resolution of all the display items is too low for publication.

- [1] <https://link.springer.com/article/10.1007/s10584-021-03132-x>
[2] <https://iopscience.iop.org/article/10.1088/1748-9326/abf964/meta>
[3] <https://www.sciencedirect.com/science/article/pii/S0360544218325039>
[4] <https://www.nature.com/articles/s41558-018-0091-3>
[5] <https://www.tandfonline.com/doi/abs/10.1080/14693062.2019.1615858>

Reviewer #2 (Remarks to the Author):

This paper is fundamentally flawed and should not be published.

These researchers do not seem to have any awareness of the biases of the models they are using and how their manipulation of the models assessing the impact of new future information perpetuates the bias against social change and societal innovation toward unrealistic technological optimism.

The authors attempt to model "the arrival of new information" as if the idea of new information in the future would be a surprise. The very set-up of the paper is problematic (and arrogant) because it assumes that the current models include "perfect information". What about all the current information (including social science and social change potential) that is not included in the IAMs?

The authors fail to even acknowledge the existing well-known and widely accepted critiques of the problematic assumptions that are perpetuated by IAMs.

The narrow focus on technologies – CDR, CCS, etc is a fundamental flaw of these models. Climate modelers fail to consider or model the possibility of systemic and structural social change and political change that could alter these pathways considerable.

The fact that modelers fail to integrate these pathways is in fact causing further delay in effective social change toward climate justice because they are essentially telling policy makers that these larger societal changes are not possible.

The authors of this article seem not to realize themselves that the assumptions currently build into the models do not adequately include current knowledge about the constraints of technological change and the potential of structural social change and systemic social innovation. The narrow perspective provided by these modelers represents a fundamental problem in climate policy and climate modeling. It is, therefore, irresponsible to publish this and perpetuate this constrained limiting perspective on what change is possible.

The conclusion of a "considerable economic challenge to achieving 1.5oC" is simplistic and misleading. Where do the authors consider the economics challenges of not achieving 1.5oC? The cost estimates included in this research area politically constructed yet the authors have no awareness of the political implications of what they are writing about.

Title – The title is confusing and lacks clarity. Given the scale of systemic and structural change that is necessary to address the climate crisis – several words in the title are unclear and confusing. "Course correction" and "Adverse information" Both of these phrases represent assumptions that are not adequately justified by the authors. The phrase course correction assumes that the world is on a certain path – and the phrase "Adverse information" does not clarify adverse to whom?

Reviewer #3 (Remarks to the Author):

This paper explores an alternative approach to constraining IAM scenarios where technological constraints are revealed only in the future, not implemented from the start of the model run. This is

arguably a more realistic way to think about how these constraints may materialise in the real world, and as such I find it an interesting contribution to the literature.

Overall, I am satisfied with the way the scenarios have been implemented and interpreted, and I find the results to be interesting, though at times a little mechanical. This is a useful contribution, especially in showing that we can react to bad news about technological availability or scale-up rates and still meet ambitious climate targets, albeit at relatively high mitigation costs.

I almost wonder if the most interesting comparison is not between the reference scenario and the adverse information scenarios, but between scenarios where we foresee adverse developments and one where we only find out about these later. It could be nice to tease out the difference between the cost of not having a technology constraint, and the cost of only finding out about this constraint later, as I think this has some more interesting policy and research implications i.e. what is the value of increasing the precision of our assumptions in modelling mitigation pathways. This I think would better capture the real contribution of the study and help to differentiate the results from other models of technology constraints.

You approach this to some extent in the first paragraph of page 18 where you talk about mitigation strategies that are robust to new adverse information. I think it would be great to draw this out a little and think about how we could think of such "hedging" strategies, and potentially how we could model them.

One other thing that struck me was that limiting scale-up rates is one of many potential ways of modelling limited technological availability. I think it would be valuable to think through and discuss some other ways this could be implemented, i.e. the discovery of negative side-effects of CDR, or issues around permanence. In some ways, it is also rather unlikely that we in 2023 (or even in 2028) receive "information" that tells us with certainty that scale up rates for renewables, or even more so for CDR technologies, are subject to a hard constraint. It would be nice to reflect on this a little more.

Finally, 2025 is now not very far in the future. I understand that what the comparison shows is the difference between modelling this new constraint from 2025 and how we have previously modelled mitigation pathways, but to the general reader this may seem a strange framing.

One small editorial comment (probably not exhaustive, though in general the paper reads very well).

page 17 line 362: (editorial) I would adjust the sentence to "This reaffirms the message from the literature on the "cost of delay" of mitigation action", and probably add a couple of citations.

Response to reviewers

Reviewer #1 comment	Authors' response
The authors explore how global cost-optimal mitigation scenarios, complying with a 1.5C target, change upon the arrival of new adverse information in either 2023 or 2028 to achieve the same climate target. This work touches upon a critical blindspot of IAMs, i.e. the models that generate that kind of scenarios: the lack of adaptiveness and/or robustness of those scenarios to new information, which inevitably will impact their optimality and feasibility. Although the study has the potential to be relevant, significant limitations must be taken care of.	We thank the reviewer for these detailed and inciteful comments, all of which we've taken on board. We think this has significantly strengthened the paper and its messaging and hope the reviewer agrees.
First, the term "course correcting" in the title and throughout the paper seems a bit of a stretch with respect to the extent of the analysis. The "correcting" is a 1-time response to a quite limited set of scenarios, either in 2025 or 2030, over the whole century. Figure 1 shows the classic dynamic adaptive pathways, but the original concept is much more dynamic. Furthermore, a paper [1] that tries to do so for a simple IAM, which I think is relevant, is not cited. Possible suggestions are to change the "course correcting" expression to something less ambitious, to explain the implications of not considering further correction points, and to discuss the acceptability of such simplification.	Understood. We cite [1] and note its relevance, whilst also pointing out the different focus and hence value-added of our study, in terms of the multiple aspects of new technological information that could be received, and that could require technology-specific responses. We are happy to de-emphasise the "correcting" nature of the scenario. We now re-name the paper "Adjusting mitigation pathways" and demonstrate that here we only show a one-time adjustment. We make clear in the Introduction, Methods and Discussion the illustrative nature of this, in light of the fact that under a truly dynamic system, multiple such adjustments could be made.
Second, this is a one-model study, and as such, it needs to address the problem of representativeness and neglected uncertainty. I recommend taking the extra effort to locate the TIAM-Grantham model within the landscape of available models, especially in terms of technological preferences and other diagnostics [2,3], for example with respect to what can be found in the last IPCC AR6 scenario database. Also note that other studies have looked at scenarios with delayed action and limited technologies in the past [4,5, for example]. A comparison with previous results and highlighting of the novelties seem to be missing here.	We accept this valid criticism of the study, which tends to occur in many single-model studies. To some extent we tried to mitigate this via situating many of the results in those from other studies including in the AR6 database (see Supplementary Information). However, we've now gone further and more fully and explicitly discussed the nature of model, in terms of both model diagnostics and also the resulting variables from the model. For the latter we use [2] primarily, since our model was involved in that paper so offers comparability in terms of diagnostics. We have also added in SI some further charts showing key technologies in our reference 1.5C scenario compared to the AR6 database of similar ("C1") scenarios with low or no

	overshoot of 1.5C, which is what our reference 1.5C (“1p5_Ref”) scenario achieves (see Supplementary Figure 8). This demonstrates that for a range of variables relevant to this study, including renewables, CCS and DAC penetration, as well as temperature profile and carbon prices, the TIAM model used in this study is in general within the range of other modelled studies. We already cite [4] and [5], but have reviewed our discussion of results in light of ref [5] and added a further new reference on the costs of delay.
Third, among the "clear policy implications" listed at the end, the importance of a diverse technology portfolio is mentioned. Nonetheless, it is unclear how the analysis supports this claim. Qualitatively, this is trivial from the assumption that if adverse things can happen to any technology, a diverse portfolio becomes an insurance. The value added here would be the whole quantification exercise, which is again hindered by being a 1-model study, for which "technology-specific outcomes should be treated with caution". Make sure to garner all the evidence you have or need to prove that a mixed-tech scenario is indeed beneficial when considering all the potential course corrections, and possibly show what the costs are of not doing so.	Noted. We have addressed this comment in two ways:  • First, our discussion is more technology-specific now, highlighting that failures in a particular technology / technology group mean that the model relies on other technologies to try to meet the 1.5C target. This allows us to highlight the specific technologies that should also be considered in terms of preparing them for rapid deployment. • Secondly, we highlight more explicitly that, through the “all adverse information” scenarios, we are able to demonstrate what the costs of technology failure are, if other technologies are not available to fill the gap.
Fourth, the adverse information is conclusive about the possibility of relying or not on a technology, or its extent. Is this a reasonable assumption? Please extract from the relevant cited literature the key points that make it credible to have such dichotomic scenarios. Also, why considering a DAC scenario even if it seems to play a relatively minor role, according to Fig 4? And why does the LimBundle LD have the same or more electricity generation than the ref, even if it is a low-demand scenario?	This is a very good point, also made by Reviewer #3. We are more explicit now (in Discussion and Methods) about the fact that our scenarios are deliberately stylised in nature, whereas in reality there would be continuously evolving information and perhaps also upward revisions of growth rates or overall technology prospects. We designed a limited role for DAC into the scenarios from the outset, without any conception of how large a role it would play, but note that it plays a relatively small role compared to some other 1.5C scenarios, as we wanted to take a more precautionary approach to technologies that are still untested at large scales.

	Our low demand scenarios sees limits to CDR and CCS, which drives the rate of electrification up to achieve the prescribed carbon budget. We have added a note to this effect in the main text.
Fifth, the model provides results for any course correction combination, no matter the delay or the tech limit. But are these results feasible at all? Otherwise, this would provide a false sense of serenity that a solution always exists. Are the high numbers in the last row of figure 9 reasonable (e.g. compared to historical statistics and/or AR6DB)? Even the ocean becomes a significant energy source (Fig 6), which is not common in this literature.	These are all valid concerns which we have already to some extent taken on. We discuss the overall narrow feasibility in terms of the implications on carbon prices. We have also added text in the discussion on the very fragile nature of the 1.5°C target at this time. We have noted that our ocean scenarios are optimistic compared to other literature and have elaborated on this. They are certainly feasible with regard to technical potential, but very ambitious compared to IRENA scenarios.
Lastly, I would improve the readability of some graphs, e.g. in Fig 2 it is almost impossible to track which line is which scenario; in Fig 3 the points are very sparse and hard to distinguish; in Fig 8b only 4 lines are distinctly visible, yet 6 are in the legend. Furthermore, the resolution of all the display items is too low for publication.	Thanks – we have added a note about the overlapping lines in Figure 8, and in any case we have produced high resolution files for each figure with our re-submission, to add clarity.

Reviewer 1 references:

- [1] <https://link.springer.com/article/10.1007/s10584-021-03132-x>
- [2] <https://iopscience.iop.org/article/10.1088/1748-9326/abf964/meta>
- [3] <https://www.sciencedirect.com/science/article/pii/S0360544218325039>
- [4] <https://www.nature.com/articles/s41558-018-0091-3>
- [5] <https://www.tandfonline.com/doi/abs/10.1080/14693062.2019.1615858>

Reviewer #2 comment	Authors' response
This paper is fundamentally flawed and should not be published. These researchers do not seem to have any awareness of the biases of the models they are using and how their manipulation of the models assessing the impact of new future information perpetuates the bias against social change and societal innovation toward unrealistic technological optimism.	We didn't set out to write a paper that argued for a purely techno-centric view of the decarbonisation challenge. That's why we include some considerable energy demand reduction scenarios. We understand and accept that the transition to a low-carbon future will require very large societal and political changes as well. Even with a focus on technologies, the paper highlights the size of the challenge, and we actually more explicitly note the societal

.	challenges that this implies. Models are necessary abstractions and their results require communication appropriately and we've tried to more explicitly set out that we only use one analytical framework here, and that further analysis, including around deeper economic, societal and behaviour changes, is advisable.
The authors attempt to model "the arrival of new information" as if the idea of new information in the future would be a surprise. The very set-up of the paper is problematic (and arrogant) because it assumes that the current models include "perfect information". What about all the current information (including social science and social change potential) that is not included in the IAMs?	We disagree with this view. Within a clearly-stated framework of perfect foresight, it is possible to usefully explore the consequences of new technological information arriving. This doesn't preclude the incorporation in the initial model runs, or in later model runs, of new societal information. In fact our low energy demand scenarios do simulate the latter to some extent.
The authors fail to even acknowledge the existing well-known and widely accepted critiques of the problematic assumptions that are perpetuated by IAMs.	We are undertaking a fairly closed-system experiment with an IAM here, rather than claiming this is the definitive or optimal pathway to net-zero. As such, whilst there is a broader critique of IAMs, we still argue that their use in such experiments is interesting and policy-relevant.
The narrow focus on technologies – CDR, CCS, etc is a fundamental flaw of these models.	We refer the reviewer to our responses above. Also, by explicitly considering the implications of limits to CCS and CDR deployment, we are contributing to the discussion around what other strategies can replace CCS/CDR reliance in these models.
Climate modelers fail to consider or model the possibility of systemic and structural social change and political change that could alter these pathways considerable.	We refer the reviewer to our responses above.
The fact that modelers fail to integrate these pathways is in fact causing further delay in effective social change toward climate justice because they are essentially telling policy makers that these larger societal changes are not possible.	It's possible this could be the case, though we note there is growing discussion around such scenarios of more radical social changes. We would argue that not all modelling exercises need to incorporate these, however.
The authors of this article seem not to realize themselves that the assumptions currently build into the models do not adequately include current knowledge about the constraints of technological change and the potential of structural social change and systemic social	We refer the reviewer to our responses above.

innovation. The narrow perspective provided by these modelers represents a fundamental problem in climate policy and climate modeling. It is, therefore, irresponsible to publish this and perpetuate this constrained limiting perspective on what change is possible	
The conclusion of a “considerable economic challenge to achieving 1.5oC” is simplistic and misleading. Where do the authors consider the economics challenges of not achieving 1.5oC? The cost estimates included in this research area politically constructed yet the authors have no awareness of the political implications of what they are writing about.	We don't think the cost, deployment or availability details of the technologies are politically constructed. We are undertaking a hypothetical experiment about what technological pathways might be followed if initial pathways are disrupted or fail to materialise. Situated within the correct caveats (which we now feel we have done), the analysis should add a useful angle to the evidence base on planning for the future in an adaptive way, both in terms of technology and social change.
Title – The title is confusing and lacks clarity. Given the scale of systemic and structural change that is necessary to address the climate crisis – several words in the title are unclear and confusing. “Course correction” and “Adverse information” Both of these phrases represent assumptions that are not adequately justified by the authors. The phrase course correction assumes that the world is on a certain path – and the phrase “Adverse information” does not clarify adverse to whom?	Thanks – the title now reads “Adjusting mitigation pathways in light of adverse new information.”

Reviewer #3 comment	Authors' response
This paper explores an alternative approach to constraining IAM scenarios where technological constraints are revealed only in the future, not implemented from the start of the model run. This is arguably a more realistic way to think about how these constraints may materialise in the real world, and as such I find it an interesting contribution to the literature. Overall, I am satisfied with the way the scenarios have been implemented and interpreted, and I find the results to be interesting, though at times a little mechanical. This is a useful contribution, especially in showing that we can react to bad news about technological availability or scale-up rates and still meet ambitious climate targets, albeit at relatively high mitigation costs.	Many thanks – we hoped our study would add a new and important angle to the existing low-carbon pathways literature, so pleased to see you think this is the case.
I almost wonder if the most interesting comparison is not between the reference scenario and the adverse information scenarios, but between scenarios where we	Our principal scenario design was to compare a scenario (1p5_ref) in which there is no serious technology constraint and then finding out about constraints later (in the 2025 and 2030

foresee adverse developments and one where we only find out about these later. It could be nice to tease out the difference between the cost of not having a technology constraint, and the cost of only finding out about this constraint later, as I think this has some more interesting policy and research implications i.e. what is the value of increasing the precision of our assumptions in modelling mitigation pathways. This I think would better capture the real contribution of the study and help to differentiate the results from other models of technology constraints.	mitigation pathway adjustment scenarios). So in essence we feel we've brought this difference out, but please let us know if there are specific additional emphases we should have given to this difference. We understand your point about showing scenarios in which we foresee adverse developments versus ones where we find out about these later. To a large extent the comparison of the 2025 pathway adjustment scenarios (following the arrival of adverse information at the end of 2023) with the 2030 pathway adjustment scenarios give us this comparison. But to further capture this, we have produced a new scenario, in which the adverse information is assumed right from the start of the globally coordinated mitigation action (in 2023). This has (as expected) a lower carbon price and system cost than the scenarios in which the pathway is adjusted later in the decade, as would be expected, given that less of the carbon budget has been consumed by the point the information arrives and is accounted for.
You approach this to some extent in the first paragraph of page 18 where you talk about mitigation strategies that are robust to new adverse information. I think it would be great to draw this out a little and think about how we could think of such "hedging" strategies, and potentially how we could model them.	This is an important suggestion, and we now add some text around how the scenarios used here could be built on to consider multiple new pieces of information over time, as part of a least vulnerability, robust decision-making exercise.
One other thing that struck me was that limiting scale-up rates is one of many potential ways of modelling limited technological availability. I think it would be valuable to think through and discuss some other ways this could be implemented, i.e. the discovery of negative side-effects of CDR, or issues around permanence. In some ways, it is also rather unlikely that we in 2023 (or even in 2028) receive "information" that tells us with certainty that scale up rates for renewables, or even more so for CDR technologies, are subject to a hard constraint. It would be nice to reflect on this a little more.	Good point and we do indeed do this now, by elaborating on the ways in which new information about limited CDR, for example, could become available, and also caveating our scenario design in the context that in reality we are more likely to receive this information in a continuously evolving manner. We use the real-world example of negative prospects of CCS at the end of the 2010s being somewhat revised in light of new investments and policies in recent months.
Finally, 2025 is now not very far in the future. I understand that what the comparison shows is the difference between modelling this new constraint from 2025 and how we have	We agree that with 2025 being close, it could make the design seem a little strange. We have discussed this point explicitly, stating that, with the GST happening in 2023, it's our expectation

previously modelled mitigation pathways, but to the general reader this may seem a strange framing.	that mitigation planning is cognizant of very near-term (as well as further in the future) new information.
One small editorial comment (probably not exhaustive, though in general the paper reads very well). - page 17 line 362: (editorial) I would adjust the sentence to "This reaffirms the message from the literature on the "cost of delay" of mitigation action", and probably add a couple of citations.	Agreed, and in response to Reviewer 1's helpful suggested citation on cost of delay, we've added this citation and another.

REVIEWERS' COMMENTS

Reviewer #1 (Remarks to the Author):

The authors have provided reasonable answers to all the points I raised. The changes they made to their manuscript include a more fitting title, a more thoughtful and technology-specific discussion of results, a more exhaustive connection with the available literature, a more transparent placement of the model in the modelling landscape, and a greater amount of explanatory notes.

My only remaining minor remark regards the readability of Figures 2 and 3. 1p5_limBundle_25 and 1p5_limBundleLD_25 seem to have the same symbols in the legend of Fig.3 (same for _30). In general, it would help to have somewhere (e.g. in Zenodo/Supplementary) also tables with the numeric data underlying the graphs to compensate for the hard readability of overlapping lines and points.

Apart from this, I recommend proceeding with publication.

Reviewer #3 (Remarks to the Author):

Overall I am broadly happy that the authors have addressed my comments from the first review. I would be happy to see this published.

One editorial suggestion.

Line 442: Should this read "are NOT limited"? or do you mean by "individual", "if ONLY single technologies are limited"?

If I may add one minor further suggestion. The rationale for presenting the bar chart in figure 4 on a polar axis is not immediately clear to me. Unless there is a particular reason for this, I would suggest this would be easier to read as a straightforward bar chart.

Response to reviewers

Reviewer #1 comment	Authors' response
The authors have provided reasonable answers to all the points I raised. The changes they made to their manuscript include a more fitting title, a more thoughtful and technology-specific discussion of results, a more exhaustive connection with the available literature, a more transparent placement of the model in the modelling landscape, and a greater amount of explanatory notes.	Thanks very much for your thorough and insightful review.
My only remaining minor remark regards the readability of Figures 2 and 3. 1p5_limBundle_25 and 1p5_limBundleLD_25 seem to have the same symbols in the legend of Fig.3 (same for _30). In general, it would help to have somewhere (e.g. in Zenodo/Supplementary) also tables with the numeric data underlying the graphs to compensate for the hard readability of overlapping lines and points.	We've now adjusted the colour scheme so hopefully this is clearer. Our hi-res figures should help too. We will indeed do this, on Zenodo.
Apart from this, I recommend proceeding with publication.	We really appreciate your constructive feedback.

Reviewer #3 comment	Authors' response
Overall I am broadly happy that the authors have addressed my comments from the first review. I would be happy to see this published.	Thanks again for your valuable review.
One editorial suggestion. Line 442: Should this read "are NOT limited"? or do you mean by "individual", "if ONLY single technologies are limited"?	Thanks for spotting this potential confusion - we've gone with your suggestion on saying "if ONLY single technologies are limited".
If I may add one minor further suggestion. The rationale for presenting the bar chart in figure 4 on a polar axis is not immediately clear to me. Unless there is a particular reason for this, I would suggest this would be easier to read as a straightforward bar chart.	No particular reason for the radar plot, other than to fit in lots of scenario data points. We've experimented with the proposed vertical bar chart format and agree this looks clearer. Many thanks for this suggestion.